

# Extinction risk of narrowly distributed species of seed plants in Brazil due to habitat loss and climate change

José Maria Cardoso da Silva[1], Alessandro Rapini[2], Luis Cláudio F. Barbosa[3] and Roger R. Torres[4]

[1] Department of Geography and Regional Studies, University of Miami, Coral Gables, FL, United States of America
[2] Departamento de Ciências Biológicas, Universidade Estadual de Feira de Santana, Feira de Santana, Bahia, Brazil
[3] Conservation International do Brasil, Belém, Pará, Brazil
[4] Natural Resources Institute, Universidade Federal de Itajubá, Itajubá, Minas Gerais, Brazil

Corresponding author
José Maria Cardoso da Silva,
jcsilva@miami.edu

## ABSTRACT

In a world where changes in land cover and climate happen faster than ever due to the expansion of human activities, narrowly distributed species are predicted to be the first to go extinct. Studies projecting species extinction in tropical regions consider either habitat loss or climate change as drivers of biodiversity loss but rarely evaluate them together. Here, the contribution of these two factors to the extinction risk of narrowly distributed species (with ranges smaller than 10,000 km$^2$) of seed plants endemic to a fifth-order watershed in Brazil (microendemics) is assessed. We estimated the Regional Climate Change Index (RCCI) of these watersheds (areas with microendemics) and projected three scenarios of land use up to the year 2100 based on the average annual rates of habitat loss in these watersheds from 2000 to 2014. These scenarios correspond to immediate conservation action (scenario 1), long-term conservation action (scenario 2), and no conservation action (scenario 3). In each scenario, areas with microendemics were classified into four classes: (1) areas with low risk, (2) areas threatened by habitat loss, (3) areas threatened by climate change, and (4) areas threatened by climate change and habitat loss. We found 2,354 microendemic species of seed plants in 776 areas that altogether cover 17.5% of Brazil. Almost 70% (1,597) of these species are projected to be under high extinction risk by the end of the century due to habitat loss, climate change, or both, assuming that these areas will not lose habitat in the future due to land use. However, if habitat loss in these areas continues at the prevailing annual rates, the number of threatened species is projected to increase to more than 85% (2,054). The importance of climate change and habitat loss as drivers of species extinction varies across phytogeographic domains, and this variation requires the adoption of retrospective and prospective conservation strategies that are context specific. We suggest that tropical countries, such as Brazil, should integrate biodiversity conservation and climate change policies (both mitigation and adaptation) to achieve win-win social and environmental gains while halting species extinction.

## INTRODUCTION

In a world where environmental changes occur faster than ever due to the expansion of human activities across all the continents (*Ellis et al., 2010*), the extinction of thousands of species is projected (*Lovejoy, 2017*). Despite the unfavorable prospect, extinctions can still be avoided (*Pimm et al., 2014*). Therefore, studies that identify extinction risks for diverse groups of organisms are relevant and raise public awareness about the importance of conserving species and their habitats as well as trigger policies and public investment required to tackle the problem (*Lovejoy, 2017*).

Extinction risks can be projected by evaluating species' attributes and the quality of the environment where they live. Among species' attributes, range size has been regarded as an important predictor of extinction risk (*Pimm et al., 2014*). In general, narrowly distributed species have small populations and, consequently, are also more susceptible to genetic drift and inbreeding, which cause the loss of genetic variability and fitness (*Hobohm, 2013*). They also have narrower habitat tolerance (*Kruckeberg & Rabinowitz, 1985*; *Wamelink, Goedhart & Frissel, 2014*) and are more sensitive to disturbances (*Lozada et al., 2008*), which make their survival highly dependent on habitat integrity (*Gaston & Blackburn, 2000*; *Wulff et al., 2013*; *Caesar, Grandcolas & Pellens, 2017*).

Restricted-range species are not distributed evenly across continents. Instead, they are clustered around some specific areas (*Kruckeberg & Rabinowitz, 1985*; *Pimm et al., 2014*; *Caesar, Grandcolas & Pellens, 2017*). Such areas are unique (*Crother & Murray, 2011*) and, from a conservation perspective, are irreplaceable and consequently top priorities for conservation actions (*Pressey, Johnson & Wilson, 1994*). Species have narrow distributions either because they originated in small areas and have not expanded since then or because their ranges contracted in response to past environmental changes (*Hobohm, 2013*). Hence, areas with such endemics, besides high conservation value, are also relevant from an evolutionary perspective because they are both centers of origin for young species (cradles) and centers of survival for old species (museums) (*Kier et al., 2009*).

Narrowly distributed species are found across all taxonomic groups, but they are more common among plants. Indeed, the smallest ranges of vascular plants are much smaller than the smallest ranges of birds and mammals (*Brown, Stevens & Kaufman, 1996*), possibly because plants are limited by edaphic factors (i.e., chemical, physical, and biological properties of soils), microclimate, and dispersal modes (*Kruckeberg & Rabinowitz, 1985*; *Major, 1988*). Plants are the basis upon which all other life depends. Yet, the number of plant species that have a high extinction risk is unknown. To date, from the estimated 386,000 species of plants (not including algae; The Plant List: http://www.theplantlist.org/), only 28,114 (7.3%) have been evaluated by IUCN Red List (*IUCN, 2019*). Among these species, 47.7% are threatened (i.e., they have been classified as vulnerable, endangered, or critically endangered) and 0.5% are extinct or extinct in wild (*IUCN, 2019*), but these numbers have been questioned. For instance, an assessment of a randomly selected sample of species indicated that the proportion of threatened plant species is around 20% (*Brummitt et al., 2015*) while other study estimated that the number of extinct species is more than four times the one estimated by IUCN Red List (*Humphries et al., 2019*). In general, IUCN's analyses

show that habitat loss is the most critical factor endangering plant species everywhere, but the impact of climate change has possibly been underestimated (*Loarie et al., 2008*; *Wiens, 2016*; *Fadrique et al., 2018*).

Global change scenarios project that habitat loss and climate change will be intensified until the end of the century and that, together, they have the potential to drive thousands of species to extinction (*Pimm, 2008*; *Oliver & Morecroft, 2014*). Nevertheless, studies on plant extinction risks assessing both factors together remain scarce (*Zhang et al., 2017*). Projections based on species distribution models usually require a large number of localities to produce meaningful results (*Araújo & New, 2007*). Consequently, extinction risks of narrowly distributed species, which are known from only a few localities, remain unchecked (*Zhang et al., 2017*). In this paper, we evaluate the extinction risk of narrowly distributed species of seed plants in Brazil due to habitat loss and climate change. Brazil is a good case study to apply this analytical approach because the country shelters the world's richest flora (*Forzza et al., 2012*), with 33,271 species of seed plants (Flora do Brasil 2020: http://floradobrasil.jbrj.gov.br), and has a vast territory encompassing distinctive ecological regions.

## MATERIALS & METHODS

### Identification of areas with microendemics

We evaluated the distribution of 3,272 species of seed plants that are endemic to Brazil with a range smaller than 10,000 km$^2$. This list of species was primarily based on the country's catalogue of rare plant species in Brazil (*Giulietti et al., 2009*). To complement and update it, we assessed the range of virtually all species of seed plants (except the orchids) that have been described as endemic to Brazil since 2008 according to the International Plant Name Index (www.ipni.org) and checked their taxonomy and distribution with information available on the Flora of Brazil 2020 website (http://floradobrasil.jbrj.gov.br), as well as their categories according to the official Brazilian Red List of threatened plants published by the Brazilian Government (*Brazilian Ministry of the Environment, 2014*). We did not assess Orchidaceae (350 species) because a preliminary evaluation showed that original publications or accurate data of occurrence were not available for many species of the family, and 40% of them were not yet on the Flora of Brazil 2020 list. The records of all 3,272 species were georeferenced and validated by checking locations with topographic maps and gazetteers. Imprecise locality data were discarded, and the species' ranges were corrected as necessary.

We intersected the localities of the restricted-range species with the fifth-order watersheds mapped at a scale of 1:250,000 by the Brazilian government (http://metadados. ana.gov.br/geonetwork/srv/pt/main.home) (Fig. S1) and excluded 918 species that were distributed in more than one watershed. Because we excluded species found in two or more watersheds, our method differs from the one used by *Giulietti et al. (2009)* to identify key biodiversity areas for rare plant species in Brazil. Watersheds with at least one species restricted to them were treated as areas with microendemics. We used these watersheds as operational geographic units because we are interested in identifying small

(median size = 189.3 km$^2$) natural areas in Brazil with narrowly distributed species (microendemics) of seed plants. This strategy allows us to assess the extinction risk of species that are more vulnerable to habitat loss and climate change in distinct areas and that can be, from a legal viewpoint, used for conservation planning purposes (*Galvão & Meneses, 2005*).

## Areas with microendemics and phytogeographic domains

We classified each area with microendemics in phytogeographic domains (hereafter domains) by using Brazil's domain (= biomes) map produced by the Brazilian Institute of Geography and Statistics (IBGE) at a 1:1,000,000 scale (*IBGE, 2004*) as a background, with a few changes in the boundaries of the Caatinga domain (*Silva, Leal & Tabarelli, 2017*) (Fig. S1). We used the centroid of each area with microendemics to classify it in one of the six Brazilian major domains: Amazon, Caatinga, Cerrado, Pantanal, Atlantic Forest, and Pampa.

## Climate change risk

To estimate the climate change risk for areas with microendemics, we used the Regional Climate Change Index (RCCI). The RCCI is a qualitative index that synthesizes a large number of climate model projections and identifies those regions where climate change could be more pronounced in a warmer climate by the end of the 21st century (*Giorgi, 2006*; *Torres & Marengo, 2014*). Therefore, the RCCI was not designed to predict specific changes in climate and vegetation that could take place in a given area because such changes are likely determined by a complex combination of physical, biological, and human characteristics (*Diffenbaugh & Giorgi, 2012*). Nevertheless, by providing an index by which the areas can be compared and ranked according to their likelihood of being affected by climate change, the RCCI is a useful tool for defining priority areas for studies of impact, adaptation, and vulnerability (*Torres & Marengo, 2014*; *Silva & Prasad, 2019*).

The RCCI uses monthly precipitation and surface air temperature data, simulated for the present-day climate (1961–1990) and projected to the end of this century (2071–2100). It combines the same 21 Earth System Models (ESMs) of the Coupled Model Inter-comparison Project Phase 5 (CMIP5) (http://cmip-gw.badc.rl.ac.uk) that were used for projections in the IPCC 5th Assessment Report. These models were forced into two Representative Concentration Pathways (RCP trajectories) in the year 2100 relative to preindustrial conditions: 4.5 W/m$^2$ and 8.5 W/m$^2$, which correspond roughly to $CO_2$ concentrations of 650 ppm and 1,370 ppm, respectively (*Moss et al., 2010*). The average resolution of CMIP5 ESMs outputs is 2° × 2° (latitude × longitude). Because of the low resolution, we used the NASA Earth Exchange Global Daily Downscaled Projections (NEX-GDDP) dataset, which is a set of downscaled climate scenarios derived from the CMIP5 with two of the four RCPs (RCP4.5 and RCP8.5). The NEX-GDDP dataset allows studies of climate change impacts at local to regional scales (*Thrasher et al., 2013*), with a spatial resolution of 0.25° (∼25 km).

The RCCI is based on four variables that were calculated separately for austral summer and winter seasons: a change of near-surface air temperature in relation to the global

average, a change in regional precipitation, and a change in the interannual variability of both temperature and precipitation (*Torres & Marengo, 2013*). The climate variables and statistics were computed as follows: (a) a change in climate variables was calculated for each model simulation, (b) the ensemble averages over different available models were computed, and (c) the two different forcing scenarios were averaged (*Torres & Marengo, 2013*; *Torres & Marengo, 2014*). The RCCI results were generated in a raster format. Using this index as raster dataset input and the areas with microendemics as the zones of interest, the zonal statistics tool in ArcGIS was applied to compute the average RCCI for each area with microendemics.

In Brazil, the RCCI ranges from 0 to 28 (Fig. S1), so we used the min-max normalization linear scaling transformation technique to rescale it to scores ranging from 0 to 1, with values close to 1 representing the highest climatic risk when comparing areas with microendemics. The areas with microendemics were also classified into three equal size categories according to their normalized climate change index: low ($\leq$0.33), medium (>0.33 and <0.66), and high ($\geq$0.66).

## Habitat loss

Habitat loss is defined here as every replacement of native vegetation by anthropogenic land-cover categories. We estimated the proportion of habitat loss in each area with microendemics by intersecting the map of these areas with Brazil's official land cover map for 2000 and 2014, produced by IBGE at a 1:250,000 scale (*IBGE, 2017*) (Fig. S1). In these maps, 14 land cover categories were identified: (1) artificial areas, (2) croplands, (3) pasturelands, (4) mosaics of cropland with forest, (5) silviculture, (6) forests, (7) mosaics of forest with cropland, (8) grasslands, (9) wetlands, (10) natural pasturelands, (11) mosaics of cropland with grassland, (12) continental lakes, (13) coastal lakes, and (14) natural exposed rock and soil. We considered as habitat loss all land-cover categories except forests, grasslands, natural pasturelands, wetlands, lakes, and natural exposed rock and soil.

We estimated the proportion of habitat loss in each area with microendemics between 2000 and 2014 and used the average annual rate of habitat loss between 2000 and 2014 of each area to project three land-use scenarios that represent immediate conservation action (scenario 1), long-term conservation action (scenario 2), and no conservation action (scenario 3). Scenario 1 assumes that natural vegetation remnants detected in 2014 are protected until the end of the century. Scenario 2 assumes that the current annual rates of habitat loss will not change until 2050, when all the remaining natural ecosystems will then be fully protected until 2100. Scenario 3 assumes that the current rates of habitat loss in each area with microendemics will continue until 2100 with no conservation action. In each scenario, we classified the areas with microendemics into three equal categories of habitat loss: low ($\leq$0.33), medium (>0.33 and <0.66), and high ($\geq$0.66).

## Combining habitat loss and climate change to estimate extinction risk

We created a matrix that combines the categories of climate change and habitat loss (Fig. 1) and classified areas with microendemics into four risk classes: (1) areas with low risk (with
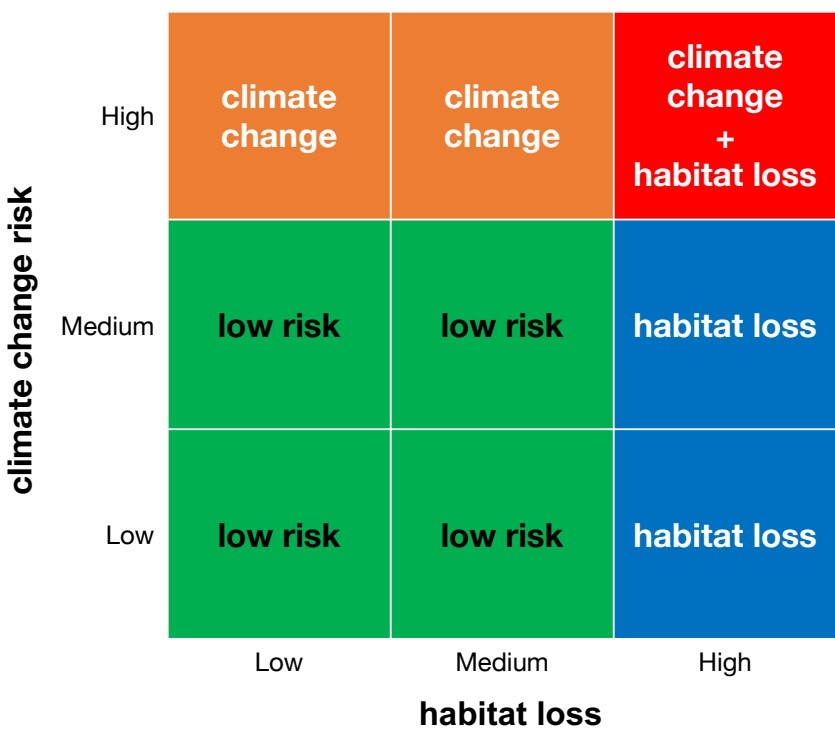

**Figure 1** Classification matrix of extinction risk for areas with microendemics and their seed plant endemic species.

low or medium climate change risk and low or medium habitat loss), (2) areas threatened by habitat loss (with high habitat loss and low or medium climate change risk), (3) areas threatened by climate change (with high climate change risk and low or medium habitat loss), and (4) areas threatened by climate change and habitat loss (with high climate change risk and high habitat loss). For each land-use scenario, we counted the number of areas and species in each risk class.

## Spatial analyses

All spatial analyses were made within a Geographic Information System (GIS) environment in ArcMap 10.6 software (Esri, Redlands, CA, USA). The final shapefile for all analyses and all maps were produced as an equal area projection (Projection: Albers Equal Area Conic; Datum: South America, 1969).

## RESULTS

We identified 776 areas in Brazil (Fig. 2A) that are home of 2,354 microendemic species of seed plants (Table S1). These areas with microendemics range from 9.1 km$^2$ to 24,885 km$^2$ (median = 898.3 km$^2$, lower quartile = 297.9 km$^2$, upper quartile = 2,183 km$^2$) and, together, cover 1,491,445 km$^2$ or 17.5% of Brazil. They are found in all six Brazilian domains (Fig. 2A), but most of them (39%) are in the Atlantic Forest. Each area shelters between one and 126 microendemics, but most of them (59%) have only one (Fig. 2A).
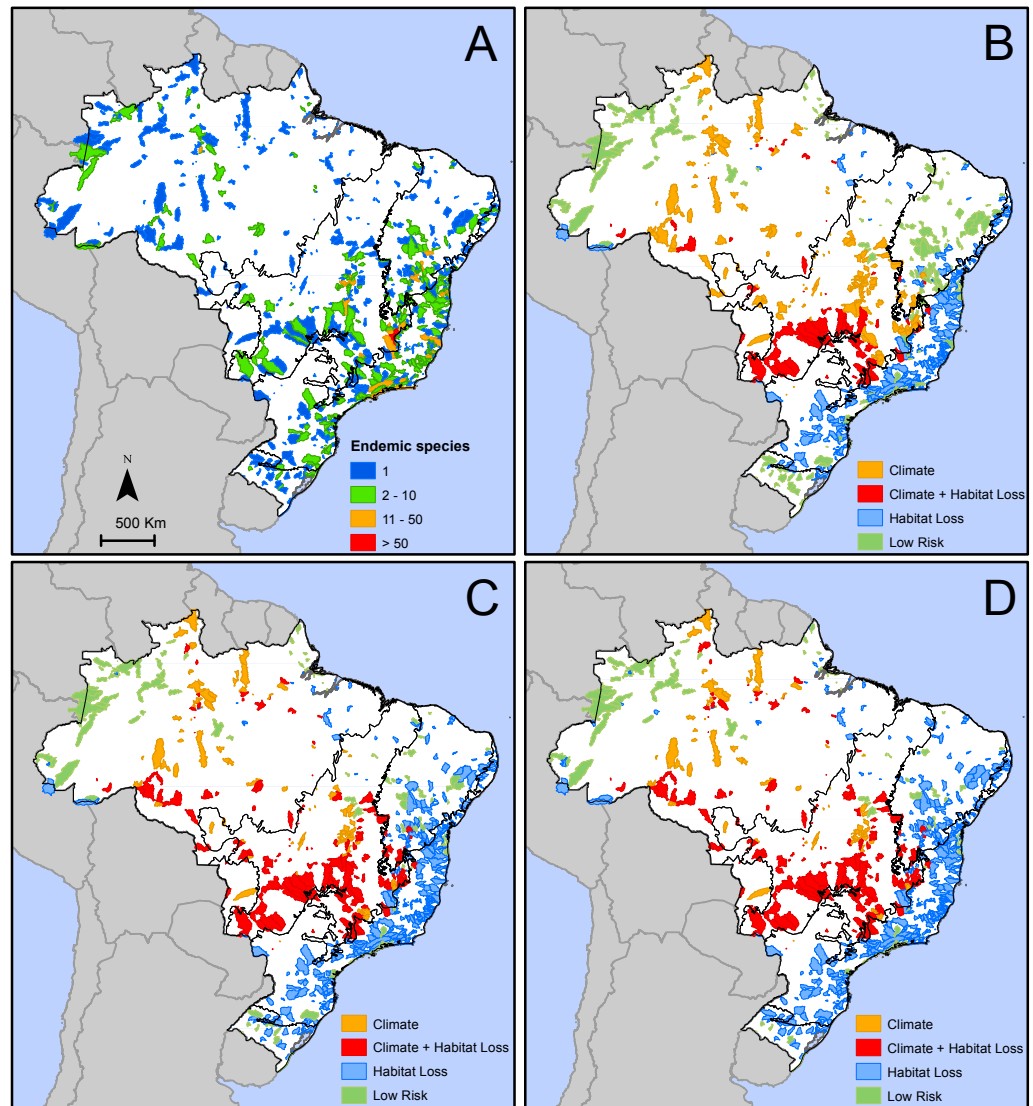

**Figure 2** Geographic distribution of areas with microendemic species of seed plants in Brazil according to (A) the number of endemic species, and (B) the classes of extinction risk in scenario 1, (C) scenario 2, and (D) scenario 3.

The areas with the highest number of microendemics are mainly found along the central Atlantic Forest and the high plateaus in Cerrado and Caatinga (Fig. 2A). Two adjacent areas in the core Espinhaço Range of Minas Gerais have more than 50 microendemics each (Fig. 2A, Data S1).

The number of areas and microendemics classified in each risk class varies according to the land-use scenario. In scenario 1, almost 70% (1,597 of the 2,354) of the microendemic species of seed plants in Brazil are under high extinction risk, 111 are in areas with both high climatic risk and high habitat loss, 926 are in areas with high habitat loss but low-to-medium climatic change risk, and 560 are in areas under high climatic risk and low-to-medium

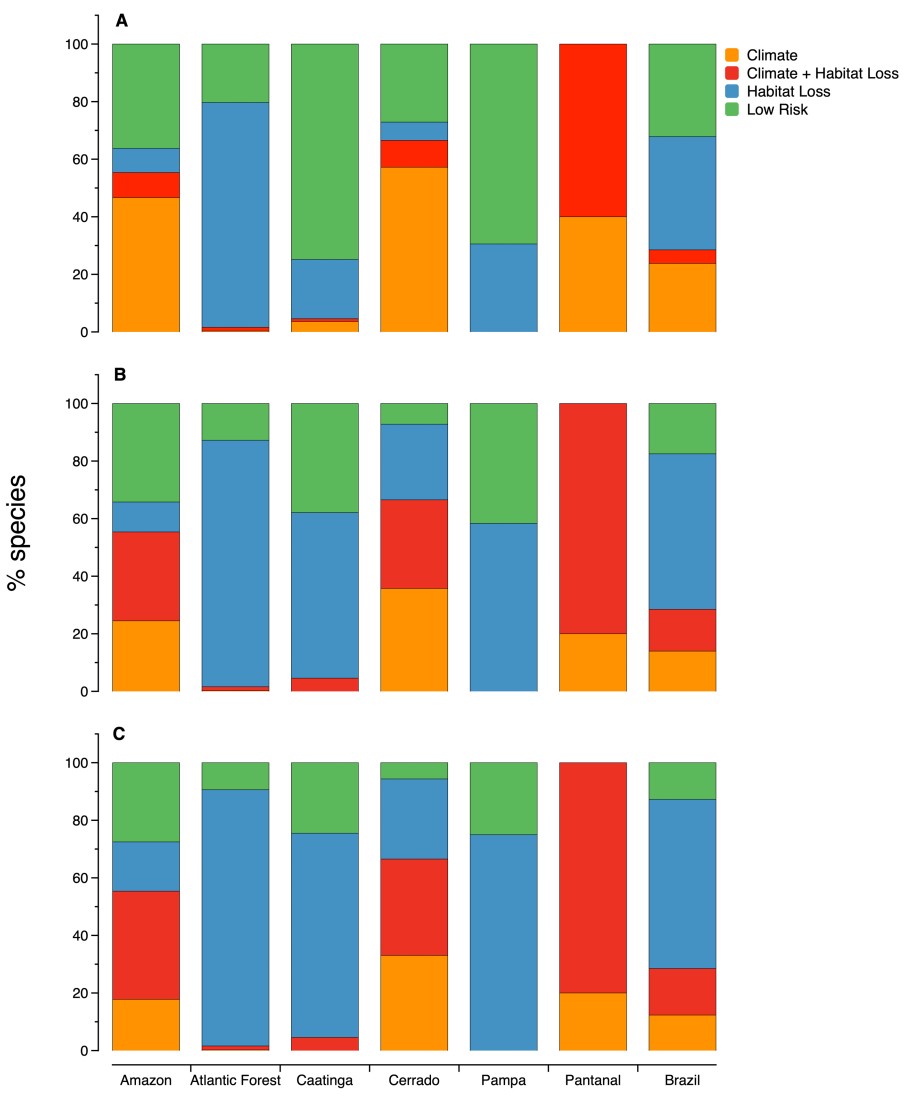

**Figure 3** The relative importance of habitat loss and climate change as threats to microendemic species of seed plants under different land-use change scenarios in the Brazilian phytogeographic domains: (A) scenario 1, (B) scenario 2, and (C) scenario 3.

habitat loss (Fig. 2B). In scenario 2, the number of threatened species increases to 1,942 (c. 82%), with 342 in areas with both high climatic risk and high habitat loss, 1,271 in areas with high habitat loss but low-to-medium climatic change risk, and 329 in areas under high climatic risk and low-to-medium habitat loss (Fig. 2C). Finally, we found that 2,054 (c. 87%) microendemics are under high risk in scenario 3, with 381 in areas with both high climatic risk and high habitat loss, 1,383 in areas with high habitat loss but low-to-medium climatic change risk, and 290 in areas under high climatic risk and low-to-medium habitat loss (Fig. 2D).

The proportion of microendemic species in each one of the four risk classes also varies across domains (Fig. 3). In all three scenarios, habitat loss is the primary factor threatening

**Table 1** Number of microendemic species of seed plants in Brazil listed and not listed on the official Brazilian Red List of threatened plant species (Brasil, 2014) according to their extinction risk classification (Fig. 1) under three different land-use scenarios.

| Scenarios | High risk | | Low risk | |
|---|---|---|---|---|
| | Red list | Not listed | Red list | Not listed |
| Immediate conservation action | 207 | 1390 | 131 | 626 |
| Long-term conservation action | 278 | 1664 | 60 | 352 |
| No conservation action | 294 | 1760 | 44 | 256 |

microendemics in the Atlantic Forest, Caatinga, and Pampa. In the Amazon and Cerrado, climate change is the primary factor threatening microendemics in scenario 1, but the combination of climate change and habitat loss is the primary threat driver in scenarios 2 and 3 (Fig. 3). In Pantanal, the species under risk are mainly affected by the combination of habitat loss and climate change. In scenario 1, the number of species under risk is higher than the number of species with low risk, except in Caatinga and Pampa (Fig. 3). However, in scenarios 2 and 3, the number of species under risk is always higher than the number of species with low risk across all domains (Fig. 3).

Among the 2,354 microendemic species analyzed in this paper, only 338 (14.3%) have been included in the official Brazilian plant Red List (*Brazilian Ministry of the Environment, 2014*). As expected, most of these species, ranging from 61.2% in scenario 1 to 86.9% in scenario 3, are also classified as having high extinction risk in our analyses (Table 1). If we count the number of microendemics that are not on the official Brazilian Red List but that are projected to have high extinction risk, we find the following numbers: scenario 1, 1,391 species; scenario 2, 1,665 species; and scenario 3, 2,017 species.

## DISCUSSION

According to the land-use scenario, the number of microendemics threatened by habitat loss, climate change, or by both factors in Brazil is between 2.86 and 3.68 times higher than the number of Brazilian plant species classified as threatened (558) by the IUCN global plant assessment (*IUCN, 2019*) but comparable (1.32 and 1.02 times smaller) to the total number of seed plant species classified as threatened (2,113) on the official Brazilian Red List (*Brazilian Ministry of the Environment, 2014*). Although the number of high-extinction risk species is smaller than the number of species on the official Brazilian Red List, these two lists show a thin overlap (c. 8%), suggesting that current conservation assessments, both at national and global levels, underestimate the actual number of plants at extinction risk by a high margin. Moreover, our findings fit well with studies in other regions of the world, demonstrating that species with narrow distributions have not had their extinction risk adequately assessed by traditional conservation assessments (*Wulff et al., 2013*; *Ocampo-Peñuela et al., 2016*; *Caesar, Grandcolas & Pellens, 2017*).

Microendemic plants are not evenly distributed across Brazil; they seem to be concentrated in relatively small watersheds where the combination of topography, climate, and soil favors both the generation and the accumulation of endemic species over time. This pattern is similar to the ones reported in other countries with high numbers of

narrowly distributed species (*Borchsenius, 1997*; *Wulff et al., 2013*). In Brazil, such regions are mostly mountains associated with stable or heterogeneous bioclimatic domains along the Atlantic Forest (*Carnaval et al., 2014*) or with old, climatically buffered, infertile landscapes (OCBILs) in the Cerrado and Caatinga highlands (*Conceição et al., 2016*; *Silveira et al., 2016*).

Habitat loss is the predominant factor increasing the extinction risk of narrowly distributed species of seed plants in Brazil. However, climate change (alone or interacting with habitat loss) can also increase the extinction risk of a substantial number of species. Despite the uncertainties on how climate is going to change ecosystems within areas with microendemics (*Torres & Marengo, 2014*), as well as how these microendemics will react to the effects of such changes, recent studies have all indicated that plant species with small ranges and narrow habitat preferences have a high likelihood to show range retraction and consequently go extinct (*Bitencourt et al., 2016*; *Zhang et al., 2017*). We found that 5% of microendemics in this study live in areas that have lost most of their original vegetation cover and have a high climate change risk, but this percentage can increase to 16.2% by 2100 if no conservation action takes place. Low habitat coverage due to insufficient protection increases extinction risk because it reduces the adaptive capacity of species to cope with climate change (*Watson, Iwamura & Butt, 2013*; *Eigenbrod et al., 2015*). Therefore, this group of microendemics is the most threatened, and the areas where they live are top priorities for more studies and conservation intervention.

The relative contribution of habitat loss and climate change as extinction drivers of microendemics varies across domains. Habitat loss is the most critical factor in the Atlantic Forest, where most of the natural ecosystems were lost due to the replacement of natural ecosystems by agriculture fields and urban areas (*Tabarelli et al., 2005*). This coastal region has a relatively low climate change risk when compared to other regions in Brazil (*Torres & Marengo, 2014*). The risk caused by climate change is higher in areas with microendemics in the Amazon, in the high plateaus in the Cerrado and Caatinga, and in Pantanal. These areas show high climate change risk through different GHG forcing scenarios and Earth System Model datasets (*Torres & Marengo, 2014*) and are still relatively intact, although habitat loss is projected to increase in these areas until 2100. Habitat loss and climate change can drive the extinction of microendemics in areas of the southwestern Atlantic Forest, the southern Cerrado, and along the Amazon's deforestation arch. Most of these areas have recently been modified by human activities as a consequence of the ongoing expansion process of large-scale commercial agriculture (*Beuchle et al., 2015*).

Projecting the extinction risk of narrowly distributed species due to habitat loss and climate change has been limited to date because they are known only from a few localities, which makes the use of species distribution models unreliable. Although our analysis addressed some of these shortcomings by shifting the focus from species ranges to the areas where these species live, our projections have limitations similar to most studies using taxonomic and distribution datasets in tropical regions. The first limitation is the reliance on information on the taxonomy and distribution of species to identify areas with microendemics. However, by relying on the previous collective effort of botanists to identify and map rare species of seed plants in Brazil (*Giulietti et al., 2009*) and counting

on electronic journals and databases to update the original dataset, we have been able to overcome some of these challenges. Nevertheless, our analysis, such as any other evidence-based projection, should be updated and evaluated over time. A second limitation of our approach is that it does not lead to a deep understanding of the complex mechanisms by which habitat loss and climate change together or separately would increase the extinction risk of narrowly distributed species. Mechanistic studies are more reliable when they are conducted on small rather than large scales (*Oliver & Morecroft, 2014*). By pinpointing the areas with plant microendemics coupled with habitat loss and climate change estimates, we hope to help researchers who are interested in extinction risks and conservation make better site selections for experiments. Finally, the third limitation of our study is that it assessed two extinction drivers (habitat loss and climate change). Thus, the impacts of drivers such as invasive species, pollution, and overexploitation on narrowly distributed native plants continue to be unchecked (*Almeida et al., 2015*; *Scarano & Silva, 2018*). Despite all the limitations above, the analytical approach used here is simple and, at the same time, robust and replicable enough to help conservationists and decision makers in tropical countries to detect essential areas for conservation and assess the extinction risks of narrowly distributed species.

Most conservation policies in Brazil and other tropical countries do not include considerations of climate change (*Kasecker et al., 2018*; *Silva & Prasad, 2019*). Consequently, they are mostly retrospective; that is, they seek to use the historical conditions of the natural ecosystems as benchmarks (*Magness et al., 2011*). Because climate change is irreversible, current policies should also be prospective. In other words, they should work proactively to facilitate the transition of the species and ecosystems to a new set of climatic conditions (*Magness et al., 2011*). The decision on what strategy to use depends on the current state of habitat loss and climate change risk of the places where species live. In general, retrospective strategies are suitable for those species endemic to areas with low risk or that have habitat loss as a major threat. For species endemic to areas with low risk, conservation through protected areas is the most effective action. For species endemic to areas where habitat loss is the primary threat, the efforts should be focused on conserving the remaining natural vegetation as much as possible through establishing protected areas and restoring vegetation in critical places.

Prospective conservation strategies are recommended for species endemic to areas that are threatened exclusively by climate change or by a combination of climate change and habitat loss. Hence, to protect species endemic to areas where the major threat is climate change, governments should establish conservation networks inside and beyond the areas with microendemics that facilitate the natural adaptation of the species and their habitats to the new climatic conditions. There are still space and time to design and implement new climate-resilient conservation networks to protect these species because such areas remain relatively intact. In contrast, for species endemic to areas that are simultaneously affected by habitat loss and climate change, governments should deploy more complex management solutions to facilitate their maintenance under new climate conditions in a landscape dominated mostly by human-made ecosystems. Such solutions should include a mix of protected areas, ecological restoration in critical sites and corridors,

active manipulation of habitat conditions to better match species requirements, and *ex-situ* conservation techniques, such as cryopreservation, seed banking, tissue culture, and cultivation collections (*Havens et al., 2006*).

The adoption of retrospective and prospective conservation strategies demands governance mechanisms that integrate conservation and national climate change strategies into a coherent policy framework. However, despite all the global efforts to promote policy coherence and integration, the policies regarding biodiversity conservation and climate change at the national scale continue to be drifting apart. We suggest that in tropical countries such as Brazil, where resources are scarce and sound environmental governance continues to be an issue, integrating biodiversity conservation and climate change policies (both mitigation and adaptation) is the most effective solution to achieve win-win social and environmental solutions while halting species extinction.

## CONCLUSION

We documented 2,354 narrowly distributed species of seed plants in 776 areas that altogether cover 17.5% of Brazil. From 70% to 85% of these species are projected to face high extinction risk due to habitat loss and/or climate change. The importance of climate change and habitat loss as drivers of species extinction varies across domains, and this variation demands the adoption of retrospective and prospective conservation strategies. At the national level, we suggest that the integration and coherence of biodiversity conservation and climate change policies is a critical step to safeguard an irreplaceable portion of Brazilian biodiversity.

## ACKNOWLEDGEMENTS

We thank Plínio Reis Moreira for putting together the data from the plant species from Brazil published since 2008 as well as Jasmine James, Thomas Lovejoy, and an anonymous reviewer who provided very useful comments on the first draft of the manuscript.

### Funding

This work was supported by the University of Miami, Swift Action Fund, and the Conselho Nacional de Desenvolvimento Científico e Tecnológico (CNPq; Grant No. 309777/2015-1). The funders had no role in study design, data collection and analysis, decision to publish, or preparation of the manuscript.

### Grant Disclosures

The following grant information was disclosed by the authors:
University of Miami, Swift Action Fund.
CNPq: 309777/2015-1.

### Competing Interests

José Maria Cardoso da Silva is an Academic Editor for PeerJ.

## Author Contributions

- José Maria Cardoso da Silva and Alessandro Rapini conceived and designed the experiments, performed the experiments, analyzed the data, contributed reagents/materials/analysis tools, prepared figures and/or tables, authored or reviewed drafts of the paper, approved the final draft.
- Luis Cláudio F. Barbosa performed the experiments, analyzed the data, contributed reagents/materials/analysis tools, prepared figures and/or tables, approved the final draft.
- Roger R. Torres performed the experiments, analyzed the data, contributed reagents/materials/analysis tools, authored or reviewed drafts of the paper, approved the final draft.

## Data Availability

The raw data is available as Table S1.

## Supplemental Information

Supplemental information for this article can be found online at http://dx.doi.org/10.7717/peerj.7333#supplemental-information.

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
