# Peer review of "Extinction risk of narrowly distributed species of seed plants in Brazil due to habitat loss and climate change"

_PeerJ, doi:10.7717/peerj.7333_

## Round 0.1 · original submission · Minor Revisions

· Academic Editor

Minor Revisions

Reviews for this manuscript are largely positive. The manuscript is well-written, the objectives are clear and major methodological flaws were not identified. Reviewer 1 has raised some pertinent points that could improve the manuscript further. In particular, I agree that the authors should include more information in the supplementary table so that readers can better related this information to the appropriate watersheds named and the figures provided. It would also be good to provide some justification for the exclusion of orchids - they are after all one of the most speciose groups of plants with countless species currently threatened or at risk due to limited ranges (fungal or orchid, or both). With respect to Figure 1, I tend to agree that it is not necessarily adding much to the manuscript, but I have an alternative suggestion. Perhaps panel A from Figure 1 can be combined with Figure 3 to provide context for the areas and watersheds, then the rest of Figure 1 could be removed as per Reviewer 1's suggestion.

Reviewer 1 ·

Basic reporting

Overall, the paper offers interesting insight into plant threats and conservation in Brazil, and is a timely review of the information found in Rapini´s book (rare plants of Brazil). In terms of balance, I think the 4 maps offered in figure 1 are perhaps not necessary (available elsewhere, not original, not part of their work).

The paper would benefit if the discussion was shortened (I found it overlong and repetitive).

In the introduction (and subsequent discussion) I found that some important literature and statements were missing. For instance, authors expand into IUCN but do not cite: https://journals.plos.org/plosone/article?id=10.1371/journal.pone.0135152
or https://www.nature.com/articles/nplants2015142.
I explain this further in my notes for them (General comments for the authors).

Thus a little more reading on plant extinction would improve the introduction and discussion.

For instance, I missed some reference or mention to the actual extinction in plants (EX, EW), which I presume is not as commonly recorded as with animals would help to provide a fuller picture.

In their methods the authors don´t explain why could they not work with the orchids? A simple search yelded 428 new records (possibly species) described from 2008 until this year. It is a considerable set of data not to use without a full justification.

Discussion:
Nowhere it has been made clear whether the endemic plant areas pointed out in the Rare plants book (Giulietti et al. 2009, of which Rapini is an author) are the same or have changed since.

When the authors write ‘even’ ex-situ conservation techniques… seedbanking it gives the impression they do not have much faith in this techniques, I suggest delete the word ‘even’.

Overall I found the discussion a bit repetitive and overlong – please compress it.

Lastly, I have highlighted the text where I thought it could be improved.

Experimental design

Results:

The biggest problem with reviewing this paper is that, while trying to check an area (by using supplementary material excell table with area data (state, area_name, ecological region) I found no link between that and the map provided (fig 3).

It would be much more useful if the name of the watershed was given in this table.

The authors mention Serra do Cipó and Diamantina plateau, but these are not named in the table. In order for me to be able to check their results during the review, and because it would make the paper much more useful, the table should also have the name of the taxa that are presumed to be endemic per area, so that this could be used by conservation practitioners.

Validity of the findings

no comment

Additional comments

Review for PEERJ
Extinction risk of narrowly distributed species of seed plants in Brazil due to habitat loss and climate change (#37305)


Overall, the paper offers interesting insight into plant threats and conservation in Brazil, and is a timely review of the information found in Rapini´s book (rare plants of Brazil). In terms of balance, I think the 4 maps offered in figure 1 are perhaps not necessary (biome map is avaliable in Forzza 2012, which they refer to, and again in BFG 2015 etc.), watershed map is very intricate and not very informative as it is presented, could be supplementary material, likewise for habitat loss and climate change maps (available elsewhere, not original).
The paper would benefit if the discussion was shortened (I found it overlong and repetititve).

Introduction:
Lines 89 to 95 when authors expand into IUCN and the situation with plants, I think it is very important to cite papers such as https://journals.plos.org/plosone/article?id=10.1371/journal.pone.0135152
Where the authors provide important insight in the situation of threatened plants worldwide
And also https://www.nature.com/articles/nplants2015142 where they can find out about the particular situation of cacti (many of them are endemic with narrow ranges).

I also think that reference or mention to the actual extinction in plants (EX, EW), which I presume is not as commonly recorded as with animals would help to provide a fuller picture. I think this may be an important area to explore in the introduction.

Lines 101-102 – unclear.

Material and Methods:
Line 119 – the authors don´t explain why could they not work with the orchids? A simple search yelded 428 new records (possibly species) described from 2008 until this year. It is a considerable set of data not to use.

Line 139 – the word Caatinga may not be familiar for all readers and comes without explanation. It is mentioned later (line 141) as one of the domains.

Line 141 – the phytogeographic domains listed here are the same areas referred to as
distinct ecological regions (line 110). Forzza et al. 2012 translate these satisfactorily as biomes, and I suggest this shorter term is used throughout to avoid confusion.

Line 162 – check the figure highlighted


Results:

The biggest problem with reviewing this paper is that, while trying to check an area (by using supplementary material excell table with area data (state, area_name, ecological region) I found no link between that and the map provided (fig 3). It would be much more useful if the name of the watershed was given in this table. The authors mention Serra do Cipó and Diamantina plateau, but these are not named in the table. In order for me to be able to check their results during the review, and because it would make the paper much more useful, the table should also have the name of the taxa that are presumed to be endemic per area, so that this could be used by conservation practitioners.

Discussion:

When mentioning the 437 species listed in the Brazilian Red List, and the 558 that appear in the IUCN, it would be interesting to know whether these species are the same or totally unrelated. The Brazilian Red List had over a thousand species listed as DD back in 2013/14, which reinforces your argument about the conservative approach of their methods.

Line 271 – 272, please read Goettsch et al. (suggested above for introduction), where traditional IUCN methods were used for one family across the board (global assessment), your affirmation that ‘ probably underestimated by using traditional
271 conservation assessments’ does not provide an accurate diagnostic of threat against plants may not hold. I also think that data present already in Green plants in the Red (2015, also suggested above) are relevant for this part of the discussion.

Line 283 – I think it is both habitat loss and degradation.

Lines 321 to 324 – it is not clear whether the endemic plant areas pointed out in the Rare plants book (Giulietti et al. 2009, of which Rapini is an author) are the same or have changed since.

Line 330 – not sure whether the areas pinpointed harbor thousands of narrowly endemic – from the maps I can see that the majority of the 700 + watersheds include a single micro-endemic

Lines 351-352 – unclear.

Lines 365 – when the authors write ‘even’ ex-situ conservation techniques… seedbanking it gives the impression they do not have much faith in this techniques, I suggest delete the word ‘even’.

Overall I found the discussion a bit repetitive and overlong – please compress it.

Lastly, I have highlighted text where I thought comprehension was difficult or where there were minor mistakes, in the hope that authors correct them.

Annotated reviews are not available for download in order to protect the identity of reviewers who chose to remain anonymous.

·

Basic reporting

no comment

Experimental design

no comment

Validity of the findings

no comment

Additional comments

this is very straightforward and useful. well written and clear....only three picky comments.

line 84 I think this is an inappropriate use of the term "orders of magnitude"; better to use a different pgrase
line 359 better "to" rather than "for" design....
line 390: should be irreplaceable not unreplaceable

---

## Round 0.2 · accepted · Accept

· Academic Editor

Accept

Thank you for your efforts to clearly address the reviewer concerns.